# The Interplay between Diabetes and Alzheimer’s Disease—In the Hunt for Biomarkers

**DOI:** 10.3390/ijms21082744

**Published:** 2020-04-15

**Authors:** Adriana Kubis-Kubiak, Aleksandra Dyba, Agnieszka Piwowar

**Affiliations:** 1Department of Toxicology, Faculty of Pharmacy, Wroclaw Medical University, 50367 Wroclaw, Poland; agnieszka.piwowar@umed.wroc.pl; 2Students Science Club of the Department of Toxicology, Faculty of Pharmacy, Wroclaw Medical University, 50367 Wroclaw, Poland; aleksandra.dyba@student.umed.wroc.pl

**Keywords:** diabetes mellitus, Alzheimer’s disease, biomarkers, S100 proteins

## Abstract

The brain is an organ in which energy metabolism occurs most intensively and glucose is an essential and dominant energy substrate. There have been many studies in recent years suggesting a close relationship between type 2 diabetes mellitus (T2DM) and Alzheimer’s disease (AD) as they have many pathophysiological features in common. The condition of hyperglycemia exposes brain cells to the detrimental effects of glucose, increasing protein glycation and is the cause of different non-psychiatric complications. Numerous observational studies show that not only hyperglycemia but also blood glucose levels near lower fasting limits (72 to 99 mg/dL) increase the incidence of AD, regardless of whether T2DM will develop in the future. As the comorbidity of these diseases and earlier development of AD in T2DM sufferers exist, new AD biomarkers are being sought for etiopathogenetic changes associated with early neurodegenerative processes as a result of carbohydrate disorders. The S100B protein seem to be interesting in this respect as it may be a potential candidate, especially important in early diagnostics of these diseases, given that it plays a role in both carbohydrate metabolism disorders and neurodegenerative processes. It is therefore necessary to clarify the relationship between the concentration of the S100B protein and glucose and insulin levels. This paper draws attention to a valuable research objective that may in the future contribute to a better diagnosis of early neurodegenerative changes, in particular in subjects with T2DM and may be a good basis for planning experiments related to this issue as well as a more detailed explanation of the relationship between the neuropathological disturbances and changes of glucose and insulin concentrations in the brain.

## 1. Alzheimer’s Disease vs. Type 2 Diabetes Mellitus

### 1.1. Alzheimer Disease Characteristic

Alzheimer’s disease (AD) is a neurodegenerative disease with two subtypes differentiating in etiopathogenesis. The most common is sporadic form with multifactorial etiopathogenesis and about 5% of AD cases are familiar form with genetic background. The clinical picture in its most common form is mainly with episodic memory disorders, cognitive deficits, emerging limitations in social behavior, and, in the advanced stages, problems with independent functioning in everyday life. Early symptoms are caused by synaptic dysfunction, which disrupts connectivity between nerve circuits, thus initiating successive memory loss. The gradual degeneration and death of neurons is accompanied by the accumulation of pathological proteins in the brain. There are many studies confirming the accumulation of pathological structures of proteins in the brain of Alzheimer’s victims. Inside the cells, hyperphosphorylated tau protein accumulate up to formation of neurofibrillary tangles (NFTs). The formation of deposits of insoluble amyloid β (Aβ) protein is observed extracellularly leading to the formation of senile plaques [1,2,3]. The senile plaques located in the brains of AD victims consist mainly of insoluble, densely packed, toxic forms of Aβ. It is formed from a larger molecule—the amyloid precursor protein, which is an integral membrane protein. Scientists suggest that amyloid precursor protein affects processes such as cell proliferation and differentiation, neurite growth, and synaptogenesis [4,5]. The oldest hypothesis regarding the cause of this phenomenon is the cholinergic hypothesis. It assumes the failure of signaling in the cholinergic system. It is based on the observation of the deficiency of the neurotransmitter—acetylcholine in the central nervous system. In addition, AD sufferers have shown a reduction in choline acetyltransferase activity—the enzyme responsible for: Acetycholine synthesis, an increase in butyrylcholinesterase and a decrease in acetylcholinesterase concentration. The main function of acetylocholinesterase is to regulate the conduction of nerve impulses through rapid hydrolysis of acetylcholine. Therefore, the basic treatment that increases the quality of transmission in the cholinergic system damaged by the disease is the use of drugs stimulating muscarinic and nicotinic receptors stimulating drugs; and acetylocholinesterase inhibitors. Butyrylocholinoesterase also participates in the control of neurotransmission and its significant amounts are found in senile plaques, neurofibrillary tangles, as well as in dystrophic nerve cells in the brain of AD cases, suggesting that better treatment effects can be brought by strategies based on inhibition of BuChE or both enzymes [6,7].

### 1.2. Carbohydrate Metabolism

The physiological concentration of blood glucose is maintained due to the antagonistic effects of insulin and glucagon. Cytokines, adiponectin, interleukin 1 (IL-1), interleukin 6 (IL-6) and tumor necrosis factor α (TNF-α) also affect the hormonal regulation of glucose metabolism. Hyperglycemia—too high blood glucose level—is associated with insulin resistance, metabolic syndrome or diabetes mellitus type 2 (T2DM). Elevated blood glucose affects peripheral tissues and blood vessels, mainly by a non-enzymatic glycation process, leading to pathological changes and developing micro- and macro-angiopathies such as: Diabetic neuropathy, nephropathy, vascular damage, and cardiovascular disease [8]. Carbohydrate metabolism is regulated mainly by insulin action through the stimulation of the absorption of glucose from the blood into fat and skeletal muscles. Disturbances in this regulating process lead to hyperglycemia and its interaction in the brain is a relatively new subject of scientific interest, while insulin and glucose homeostasis abnormalities can contribute to developing T2DM as well as neurodegenerative conditions in different ways [9,10].

#### 1.2.1. Glucose Metabolism in the Brain

The insulin signaling pathway plays a key role in regulating the transmembrane passage of glucose. It depends on various mediators such as glucose transporters (c glucose transporter 4, GLUT4), phosphatidylinositol (PI) 3-kinase–dependent and –independent pathways and, to a large extent, on astrocytes involved in the structure of the blood-brain barrier, significantly contributing to maintaining energy homeostasis in the brain [11,12]. Astrocytes easily transport glucose from the blood through endothelial cells and transfer metabolic energy substrates between blood and neurons. They also contain large amounts of granules filled with glycogen which are most observed in astrocytes placed in areas with high synapse density [13]. It has been shown that glycogen can be used to supply lactate for neuronal metabolism during hypoglycemia, suggesting a very important role of astrocytes in both physiological glucose utilization and hypoglycemia [14]. Glucose transporters, including GLUT 1 and 3, are very important in regulating glucose transport from blood to neurons in the brain. In addition, they seem to be important factors involved in the pathogenesis of Alzheimer’s disease (AD) [15]. Due to the limited permeability of the blood-brain barrier (BBB) and the relative lack of local brain carbohydrate stores, any abnormalities in GLUT function and expression disturb brain glucose homeostasis. Liu et al. [16] demonstrated a significantly reduced number of GLUT 1 and GLUT 3 as well as their regulator—hypoxia-inducible factor 1—in the brain of victims of AD together with a decrease in the protein O-GlcNAcylation level. They suggested that this may contribute to impaired glucose uptake and its transport to neurons and glial cells to the extent leading to impaired glucose metabolism in the brain. Both abnormal glucose metabolism and oxidative stress contribute to the formation of advanced glycation end products which in consequence induces a series of biological processes that cause further diabetic complications such as retinopathy, nephropathy, cardiomyopathy or neuropathy. The histopathological studies revealed the positive staining for advanced glycation products in proximity of Aβ plaques and NFTs. That’s why, advanced glycation products are widely accepted to be important player in the AD pathology. It was shown that glycation of Aβ and tau protein promoted by advanced glycation products, leads to Aβ aggregation and the formation of NFTs, respectively [17]. The main ways which have been a consequences of acute hyperglycemia action in the brain are present in Figure 1.

Intracellular oxidative catabolism consists of complex pathways including glycolysis and the pentose phosphate pathway in the cytoplasm as well as the Krebs cycle and oxidative phosphorylation in mitochondria. Both abnormalities in transport and intracellular oxidative catabolism affect glucose metabolism in the brain, and thus probably contribute to metabolic disorders in AD [18]. Cases with AD have glucose transport abnormalities due to insulin resistance and intracellular metabolic changes, which are the consequences of mitochondrial dysfunction [19]. Both hyper- and hypo-glycaemia have a significant impact on brain function, especially cognition, which has been shown in clinical studies [20]. Mosconi et al. [21] demonstrated a progressive reduction in glucose metabolism in the brain of AD subjects, while the location and extent of these changes were correlated with the stage of the disease. AD victims showed regional glucose hypo-metabolism in the hippocampus and the entorhinal cortex—memory-related areas In addition, the decrease in the local glucose metabolism rate also reflects regionally reduced density and synaptic activity, suggesting that selective and local reduced glucose metabolism is associated with specific cognitive impairment in AD cases [22]. Furthermore, the researchers observed that metabolism reduction also occurs in people at high risk of developing AD several years or even decades before clinical signs appear [23]. An example is the mutated presenilin 1 (PS1) gene carriers with the familiar form of AD, which show a significant reduction in the brain glucose metabolism rate by an average of 13 years before manifestation of symptoms [24]. Hiltunen et al. [25] in mouse model with overexpressing mutant amyloid precursor protein and presenilin-1 (APdE9) demonstrate that the APdE9 transgene enhanced hyperinsulinaemia and insulin resistance induced by genetic or dietary factors, which can be an additional import regulatory molecular mechanism that link T2DM and AD pathology. Additionally dyslipidemia and hypercholesterolemia, a well-known elements contributing to the pathology of T2DM, are also indicated as independent risk factors for AD. Especially, attention is paid to the Apolipoprotein E (APOE), which participates in the transport of cholesterol and lipoproteins within the circulatory system. It is indicates that risk of AD associated with the APOE ε4 allele might be exacerbated by T2DM, about 4-times [17]. Middle-aged people with normal cognitive function who are holders of two copies of the apolipoprotein Eε*4* allele were also found to have reduced glucose metabolism in the brain [26]. 

#### 1.2.2. Insulin Role in the Brain

Over the last few decades, the brain has been proven as an insulin target region. In the CNS, insulin influences i.a. glucose metabolism, memory and cognitive functions [27]. AD and impaired insulin signaling, which is characteristic of T2DM, have several common pathophysiological features including Aβ accumulation. Regardless of whether the insulin comes from the peripheral nervous system or was produced in the brain, insulin works by activating specific brain receptors and is degraded by the same insulin degrading enzyme (IDE) together with Aβ [28]. Therefore, if insulin concentration in T2DM is too high, it may act as a competitive substrate resulting in the accumulation of Aβ in the form of senile plaques. Correct insulin signaling provides a physiological defense mechanism against the toxic effects of Aβ oligomers on synapses by reducing oligomer binding sites in neurons. Insulin has multiple anti-amyloidogenic effects on human nerve cells, including preventing the translocation of an intracellular domain fragment of the amyloid precursor protein (APP) into the nucleus, increasing the transcription of anti-amyloidogenic proteins and increasing the α-secretase-dependent APP processing pathway [29]. 

Many scientists analyzed the effect of glucose or insulin on intracellular or extracellular Aβ levels [10,27,28]. An in vitro study showed that in primary neuronal cultures and in neuroN2a cells overexpressing the β chain of APP, insulin reduces the intracellular accumulation of Aβ by initiating the transport of APP/Aβ from the Golgi apparatus into the cell membrane. Thus, in addition to inhibiting IDE degradation, insulin increases the concentration of extracellular Aβ_40/42_ modulating APP transport [30]. In addition, studies revealed the role of the tyrosine/mitogen-activated protein (MAP) kinases receptor pathway in the regulation of intracellular Aβ transport [31]. Experiments on astrocyte responses to Aβ_40_, the main cellular type involved in the maintenance of synaptic glutamate concentrations, showed a decreased astroglial glutamate uptake capacity. The levels of the astrocytic glutamate transporters, glutamate transporter-1 and glutamate–aspartate transporter were decreased in Aβ_40_-treated astrocytes. The MAP kinases, extracellular signal-regulated kinase, p38 and c-Jun N-terminal kinase were activated at the beginning of the Aβ_40_ incubation and the entire pathways differentially modulated the activity and levels of glutamate transporters. Studies on C57BI/6 mice demonstrated that high glucose levels cause a significant increase in neuronal reactive nitrogen species (RNS) that leads to an increased aggregation of oligomeric forms of Aβ_42_ and a compromise in mitochondrial bioenergetics. It seems that both diseases are oxidizing, altering the redox state of cortical and hippocampal neurons in the brain. The resulting nitrosative stress causes aberrant S-nitrosylation reactions on proteins responsible for Aβ accumulation [32]. Experiments performed on human stromal vascular cells and differentiated adipocytes, which were incubated with different concentrations of glucose and insulin, showed an increase in Aβ secretion into the extracellular space. Furthermore, adipocytes incubated with Aβ had a decreased expression of insulin receptor substrate-2 and a reduced Akt-1 phosphorylation. The results obtained suggest that overlapping patterns of metabolic dysfunction may be common molecular links between these complex diseases [33].

### 1.3. Brain Energy Metabolism and AD Onset

The above described metabolic interrelations together with the studies on the metabolism of glucose transporters in the brain in people affected by AD reveal a strong correlation between the onset of the disease and glucose metabolism in the brain. The data obtained from obese and T2DM subjects suggest a gradual decrease in the proper brain response to high glucose levels. Epidemiological studies point out that hyperglycemic sufferers have a higher AD risk and demonstrate a greater conversion rate from mild cognitive impairment to AD [34]. Cerebral regions sensitive to the aggregation of Aβ and NFTs display significantly higher glucose concentration in AD. Moreover, elevated levels of brain tissue glucose are associated with greater severity of both Aβ deposition and neurofibrillary pathology [35]. In a community-based controlled study performed together with pathological study of autopsy cases from this same community, authors have found that both T2DM as well as impaired fasting glucose were twice more prevalent in AD versus non–AD control subjects. Over 80% of AD cases had either T2DM or impaired fasting glucose [36]. T2DM induces both functional and structural changes in the brain, so disturbances connected with glucose and insulin metabolism are indicated currently as the most interesting and important in these phenomenon. Abnormalities in insulin signaling, dysregulated glucose metabolism, the formation of advanced glycation end products, and secondary disturbances connected with these, are features common in T2DM but they are also indicated as potential agents significantly associated with AD. Some data shown that 80% of AD cases exhibited either impairments in glucose tolerance or frank T2DM [37,38].

On the other hand, there are data showing that cerebral pathology typical for AD is not significantly associated with glucose homeostasis disturbances [21,22]. Some authors have found no evidence of an association between T2DM and AD neuropathology, however, they observed that certain subgroups, such as APOE allele ε4 carriers, had higher odds of accumulation of neurofibrillary tangles [39] But others indicated that history of T2DM and hypertension was independently associated with a shorter lifespan of AD sufferers [40] Lately, You et al. [41] revealed that current smokers and subjects with lower income, plasma glucose levels, body mass index (BMI), and subjects with hypertension, dyslipidemia, vascular complications, depression, and under insulin treatment developed dementia more frequently. In addition, few studies have addressed the relationships between longitudinal changes in measures of glucose tolerance and AD pathology. In the prospective cohort study, serum glucose, insulin, and HOMA values and postmortem AD pathology were assessed. After statistical analysis using analysis of variance and a continuous analysis with linear mixed models It was found no association between lifetime measures of glucose homeostasis and standard measures of AD pathology or cortical fibrillary Aβ deposition [29].

## 2. The Effect of Metabolic Disturbances on Amyloid Production

### 2.1. Impact of Glucose Metabolism on Amyloidogenesis

Chronic hyperglycemia is characteristic of T2DM and extensive evidence suggests that it has a toxic effect on brain function. In a cohort of healthy, older, nondiabetic individuals without dementia, lower glycated haemoglobin (HbA1c) and glucose levels were significantly associated with better scores in delayed recall, learning ability and memory consolidation [42]. Macauley et al. [43] observed that the induction of acute hyperglycemia in 3-month-old wild APP_SWE_/PS1de9 transgenic mice directly increases the production of different forms of Aβ in the interstitial fluid, which is associated with an increase in neuronal activity. These effects are exacerbated in 18-month-old APPswe/PS1de9 mice, in which the accumulation of toxic forms of Aβ has already been noted. The scientists also showed that Adenosine triphosphate (ATP)-dependent potassium channels behave as metabolic sensors for changes in glucose levels, causing changes in neuronal excitability and indirectly contributing to extracellular Aβ deposition [43]. In other studies in AD Tg2576 mice, high fat diet-induced insulin resistance promoted an increased aggregation of Aβ_40_ and Aβ_42_ in the brain together with increased gamma-secretase and decreased IDE activities [44]. High glucose levels may increase Aβ_40_ production by inhibiting APP degradation [45]. Li et al. noticed elevated levels of AGE in the blood and brain of AD victmis [46]. Receptors for AGE (RAGE) are abundant in both microglia and neurons and are responsible for pathological consequences. Studies show their increased expression in the astrocytes and microglia of the brain of AD [47]. Aβ is a RAGE ligand, and the Aβ-RAGE interaction significantly increases neuronal stress, Aβ accumulation, impaired learning and memory and leads to the activation of pro-inflammatory cytokines (Figure 2) [48]. Alternative mechanisms may include the accumulation of autophagosomes to enhance APP cleavage or increased expression of β-secretase 1 (BACE1) [49]. It is suspected that hyperglycemia may impair cognitive function and cause the development of dementia. It can contribute to structural and functional damage to neurons and nerves in the brain, cerebral hemorrhage and increased accumulation of Aβ [50].

### 2.2. The Role of Insulin Signaling in the Amyloid β Cascade

Over the last two decades, numerous studies have shown that insulin receptors (IR) are widespread in the brain [51]. Insulin plays a vital role in regulating various central nervous system tasks, and these functions include, among others, proper functioning of synapses, activation of neuronal stem cells, growth and repair of neurites, and has an impact on learning processes and memory [52]. Therefore, dysfunction at various levels of insulin signaling and its metabolism may contribute to a number of brain disorders. Evidence points to a link between AD and abnormal insulin signaling in the CNS. They share many common pathophysiological features including impaired insulin sensitivity, Aβ accumulation, tau hyperphosphorylation, brain vasculopathy, inflammation and oxidative stress [53]. Insulin and the insulin-like growth factor (IGF) regulate a number of biological processes by binding and activating two closely related tyrosine kinase receptors: IR and the IGF-1 receptor (IGF-1R). Hölscher et al. [54] suggest that insulin has neuroprotective properties and exerts a neurotrophic effect on CNS neurons. After binding insulin to IR, the receptor is auto-phosphorylated, then the activated IR phosphorylates the cascade of IR protein substrates (IRS). One of the main further IRS pathways is the phosphatidylinositol 3 kinase cascade/protein kinase b (Akt). This, in turn, affects other signal transduction pathways, including threonine-serine protein kinase, glycogen synthase kinase 3 beta (GSK3β) and the FoxO transcription factor family. For many of these paths, a key role in the proper functioning of the brain is assigned [55]. T2DM is a disease with a dynamically changing course, progressing from dominant insulin resistance, through compensatory hyperinsulinemia, to the exhaustion of the secretory capacity of pancreatic β cells. Studies demonstrate that insulin resistance accelerates the production of Aβ, initiating its accumulation. In the case of 3xTg-AD mice (APP_SWE_, PS1_M146V_, tauP301L), in which insulin resistance was induced by applying a high-fat diet, increased levels of Aβ_40_ and Aβ_42_ in the brain and enzymes that are involved in their formation (e.g., γ-secretase) were shown [56]. The study on 5xFAD mice (APP_SWE_, APP_Florida_, APP_London_, PSEN1_M146L_, PSEN1_L286V_), in which T2DM was induced by streptozotocin, revealed that insulin deficiency changes APP processing by increasing β-site amyloid precursor protein (BACE-1) expression [57]. Another study showed that insulin resistance can alter APP processing by activating autophagy [58]. The insulin-degrading enzyme, which is zinc-thiol-dependent metalo-endopeptidase, is also responsible for the degradation of Aβ. If insulin concentration in T2DM is too high, it acts as a competitive IDE substrate and inhibits the degradation of Aβ, which gradually accumulates, forming insoluble plaques [1]. Studies indicate that IDE gene deletion in mice (IDE^−/−^) caused a 50% decrease in Aβ clearance in brain homogenates and primary neuronal cultures. In addition to hyperinsulinemia and glucose intolerance, which characterize T2DM mice lacking IDE, showed an increased accumulation of endogenous Aβ_40_ in the brain [28]. On the other hand, it is Aβ oligomers that can inhibit insulin signaling, which shows a feedback mechanism. Small Aβ oligomers contribute to synaptic toxicity and further events that lead to neurodegenerative processes. It is suggested that these processes include the effect of oligomers on insulin signaling by inhibiting IR autophosphorylation and significantly reducing the activity and levels of IR on the surface of neuronal dendrites in the brain [59]. The hippocampus, a brain region that is key to memory and learning, was found to demonstrate particularly high levels of IR, suggesting that insulin could contribute to synaptic plasticity mechanisms and memory formation. Indeed, insulin regulates neuronal survival, acts as a growth factor, and regulates circuit function and plasticity. Furthermore, intranasal insulin treatment improves memory in healthy adults, without changing blood levels of insulin or glucose [60]. Given that IR play a vital role in important processes involving learning and memory, their significant decline may be an important early mechanism underlying memory and cognitive impairment. Aβ oligomers have also been found to increase Akt phosphorylation at Ser473 residues, resulting in insulin resistance [61].

## 3. The Effect of Metabolic Disturbances on Tau Protein Metabolism

### 3.1. Impact of Glucose Metabolism on Tau Protein Aggregation

Abnormal glucose metabolism is a hallmark of T2DM, but scientists suggest that it also contributes to the aggregation of tau protein [62]. In recent years, many studies on T2DM have focused on analyzing the activation of the inflammatory signaling pathway in immune cells. It is indicated that toll-like receptors (TLRs), specifically TLR9, are involved in the regulation of innate immunity. A significant increase in neuron expression and an increase of tau hyperphosphorylation in high glucose media were observed [63]. These receptors are usually activated by a non-methylated DNA sequence CpG. It is a short sequence containing cytosine, guanine and a phosphodiester linkage. CpG is found in bacterial and eukaryotic mitochondrial DNA [64].

In addition to participation in inflammatory reactions, TLR9 can affect non-immune cell metabolism. Studies confirming this phenomenon indicate that mature neurons, in order to survive in acute hypoxia, can reduce their cell metabolism by activating the TLR9-5’AMP-activated kinase (AMPK) signaling pathway [65]. Other studies confirm an increased stimulation of the AMPK in the brain of AD cases that induces tau phosphorylation [66]. Sun et al. [63] were the first to examine mature neurons isolated from the hippocampus, detecting an increased expression of TLR9 in them, accompanied by the accumulation of hyperphosphorylated tau. This process depends on the protein kinase activated by p38 mitogen. The association of glucose metabolism and tau protein aggregation is also confirmed by a study, in which an antidiabetic drug (e.g., metformin) increases the activity of protein phosphatase 2A (PP2A), which is the enzyme responsible for the process of tau dephosphorylation. The study used neurons isolated from the brain of 8c mice (B6.Cg-Mapt tm1(EGFP)Klt Tg(MAPT)8cPdav/J), which were then incubated with metformin. It was revealed that by increasing PP2A activity, in vitro tau phosphorylation and in animal models is reduced. The obtained results demonstrate a potential role of biguanides in the prevention or treatment of AD [67].

Other processes that affect the modification and aggregation of tau protein such as acetylation and glycosylation are a relatively new object of interest to scientists. Brain cell studies in PS19/PDAPP mice and subjects with AD and related tauopathies indicate that acetylation interferes with tau binding to microtubules and thus their stabilization. Insoluble NFTs are formed, which point to acetylation being a pathological factor of tau aggregation [68]. Glycosylation involves the attachment of oligosaccharide molecules to proteins and lipids. O-glycosylation is the attachment of sugar residues to serine or threonine hydroxyl groups, while N-glycosylation involves the attachment of sugar groups to asparagine amino groups in proteins. In AD, the process of linking β-n-acetyl-d-glucosamine residues with O-glycosidic (O-GlcNAc) bonds to proteins is impaired. Post mortem studies point to a significant decrease in O-GlcNAc tau glycosylation in the brain of AD victims compared to the control [69]. In vitro studies on the nucleocytoplasmic activity of o-β-n-acetylglucosaminyl transferase (OGT) were also tested by mass and nuclear magnetic resonance spectroscopy. Its potential substrates are specific tau domains, but the available analytical methods hinder the exact location of O-GlcNAc sites in tau. Using phosphorylated peptides, attempts were made to establish a relationship between the tau phosphorylation process and O-GlcNAc. Hyperphosphorylation of serine residues (Ser396 and Ser404) was shown to be significantly reduced by Ser400 O-glycosylation [70]. The inverse relationship between these processes is confirmed by another study which indicates a significantly reduced O-GlcNAc tau protein isolated post mortem from the brain of AD sufferers compared to the control sample. Administering an OGA inhibitor (an enzyme catalyzing the hydrolytic cleavage of O-GlcNAc) in an AD model of mice points to a reduction in NFTs levels and pathological forms of tau, and slowing disease progression [71]. However, a reduction in tau phosphorylation is observed for a short time and prolonged inhibition of O-GlcNAc does not affect phosphorylation, which may be due to cell adaptation over time. These studies suggest that glycosylation exerts different effects on tau protein metabolism. It was hypothesized that abnormal T2DM-connected brain glucose metabolism may lead to a decrease in O-GlcNAc levels, which results in neuroprotective mechanism failure and triggers a cascade of pathological forms of tau resulting in AD progression. In combination with the accumulation of Aβ and the formation of NFTs, impaired glucose metabolism and hyperglycemia as well as its pathological effects form a vicious circle, jointly contributing to brain dysfunction in AD [18].

### 3.2. Improper Insulin Metabolism and Tau Protein Synthesis

Insulin and IGF-1 were shown to regulate the process of tau phosphorylation by inhibiting the key kinase phosphorylating tau protein GSK-3β in nerve cell cultures. Impaired insulin signaling in the brain leads to a decrease in Akt activity, which results in increased GSK-3β activity. This phenomenon causes tau hyperphosphorylation and, thus, a higher formation of NFTs [72]. In addition, peripheral hyperinsulinemia promotes in vivo tau phosphorylation. It was shown that in mice lacking IGF-1 and IRS-2 genes, tau phosphorylation is dramatically increased. Lack of gene for IGF-1 causes excessive GSK-3β activity and specific target points for this enzyme are phosphorylated. It seems that under physiological conditions, insulin and IGF-1 prevent tau phosphorylation in the brain [73]. Given that T2DM is characterized by insulin resistance, hyperinsulinemia and impaired insulin signaling, it is not surprising that the increased GSK-3β activity in T2DM may lead to an increase in Aβ production and increase tau phosphorylation. Decreased insulin signaling can lead to an impaired expression of the tau gene, resulting in a decrease in soluble tau levels, followed by the accumulation of hyperphosphorylated tau that causes the deformation of the neuronal skeleton, neurite collapse and abnormalities in synapses [74]. Planel et al. [75] studied C57BL/6NJcl mice with streptozotocin-induced T2DM. Those authors noticed massive tau hyperphosphorylation located in axons and neuropil cells. It prevented protein from binding to microtubules. Additionally, a significantly reduced level of 2A protein phosphatase was identified, which may also be a potential mechanism contributing to excessive tau phosphorylation. On the other hand, Rodriguez-Rodriguez et al. [76] observed that tau hyperphosphorylation can affect the development of insulin resistance. They suggest that insulin accumulates in neurons burdened with hyperphosphorylated tau protein. Cells in which insulin accumulated demonstrated signs of insulin resistance and reduced levels of insulin receptors.

## 4. Biomarkers for Alzheimer’s Disease Diagnosis

Alzheimer’s disease can be diagnosed with complete certainty after death, when microscopic examination of the brain reveals the characteristic plaques and tangles. Early sign evaluated by doctors include among others, memory impairment, difficulty concentrating, planning or problem-solving, confusion with location or passage of time, language problems and changes in mood, such as depression or other behavior and personality changes. MRI, CT and PET scans helps to rule out the other causes of psychological changes and helps to distinguish between different types of degenerative brain disease. Scientists are investigating a number of disease markers and diagnostic tests, such as genes, disease-related proteins and imaging procedures, which may accurately and reliably indicate whether you have AD and how much the disease has progressed. There is a strong need to find, evaluate and imply new, more reliable diagnostic markers that will have strong specificity and sensitivity towards AD.

Changes in the concentration of possible biomarkers in body fluids, especially in CSF, can be observed much earlier, before the onset of AD symptoms. Levels of Aβ_42_, total tau and phosphorylated tau showed a high diagnostic accuracy but were still unreliable for preclinical detection of AD [77]. As highlighted by the documents published by the U.S. Food and Drug Administration and the European Medicines Agency: “the use of biomarkers in AD diagnostics creates hope for a more accurate assessment of the stage and progression of the disease, the development of more effective therapy, and can also be a measure of treatment effectiveness” [78,79]. Searching for and studying body fluid AD biomarkers are not only useful in exploring the onset and etiopathogenesis of the disease, but can also contribute to an improved prognosis and diagnosis accuracy.

### 4.1. T2DM-Related Biomarkers in AD

The link between peripheral/brain insulin resistance and cognitive deficiency may possibly be mediated through high levels of neurotoxic molecules crossing the blood-brain barrier [18]. The findings concerning insulin signaling deficits and insulin resistance in the brain of AD sufferershave led to the creation of a new term: type 3 diabetes [80]. The question is how changes occurring during the so-called “type 3 diabetes” development can be monitored and distinguished from biomarkers characteristic of each disease separately. Hypometabolic regions in the posterior cingulate-precuneus, posterior lateral and medial temporoparietal association cortex and in the lateral frontal cortex identified via PET imaging with glucose labeled with ^18^F-based radiotracer; can be used as a metabolic biomarkers for early AD diagnosis [81]. These regions are crucial for language and working memory; therefore, their metabolic dysfunction could be used as a prognostic marker for AD vulnerability among healthy elderly as well as an AD progression indicator. Hypometabolism in the medial temporal cortex is significantly linked with early stages, whereas metabolic failure in the temporal-parietal cortex is observed together with disease advancement. Interestingly, changes in the metabolism of reduced glucose that are characteristic of the late- and early-onset form of AD are observed. As a result, reduced anaerobic activity in the posterior lateral temporoparietal region is typical of the early-onset form of AD, while a decrease of aerobic glycolysis in the medial temporal region is correlated with the late-onset form of AD [15]. In the study performed on T2DM vervet monkeys (Chlorocebus aethiops sabaeus), Kavanagh et al. [82] demonstrated that CSF glucose and plasma lactate levels were strongly correlated with Aβ_40_ and Aβ_42_ concentrations in CSF. This model transforms from normoglycemia to impaired fasting glucose and full-blown diabetic states with decreasing levels of Aβ_40_ and Aβ_42_ in CSF and increased plasma and CSF glucose levels. Furthermore, T2DM vervet monkeys demonstrate reduced levels of amino acid and acylcarnitines in CSF. The latter, byproducts of mitochondrial fatty acid, amino acid and glucose catabolism, play an important role during the fasting environment in the brain. These data suggest that peripheral metabolic disturbances have consequences manifesting themselves in perturbations in brain energy metabolism. In the study by Gupta et al. [83], higher values of glycated hemoglobin (HbA1_C_) has been shown as a good indicator of the association of dementia and cognitive dysfunction with uncontrolled T2DM. Together with poor results of the General Practitioner Assessment of Cognition, the Memory Impairment Screen and Mini COG^TM^ this data could be a good base for the assessment of dementia and cognitive decline in T2DM cases. It was also found that Aβ_42_ and PS1 levels were considerably higher in overweight and obese adolescents compared to normal-weight people [84]. One study put emphasis on the value of evaluating indices of insulin resistance and their consequences (oxidative stress and inflammation) combined with phosphorylated tau and Aβ in CSF-based multiplex assays. Additionally, hyperinsulinemia and hyperglycemia caused by insulin resistance hasten the formation of neuropathologic changes typical of AD. Preclinical and clinical studies serve as a confirmation that insulin could be beneficial to the treatment of AD as well as may be a novel biomarker in AD [85,86].

### 4.2. S100 Protein Family as a Potential Biomarkers of AD

Approx. twenty-five low-molecular (10–12 kDa) proteins belong to the family of S100 proteins, which are characterized by cell-specific and tissue-specific occurrence and a similar primary structure [87]. Proteins from the S100 family have two helix-loop-helix domains, a helix-loop-helix structural domain, with which they bind calcium ions with different affinities [88]. Some proteins belonging to the S100 family may also bind to zinc and/or copper ions [89]. S100 proteins are multifunctional, participating in, among others, the regulation of protein phosphorylation, cell growth, motility and differentiation, the cell cycle and transcription processes [90]. Proteins from the S100 family can be secreted outside the cells, indicating their extracellular activity [88]. The best characterized proteins from the S100 family with respect to extracellular activity include the S100A8 and S100A9 proteins [91,92]. S100A8 and S100A9 proteins can form the S100A8/S100A9 heterodimer, which modulates the activity of casein kinase I and II, which phosphorylate topoisomerase I and RNA polymerase I and II [93]. Intracellularly, the S100A8/S100A9 heterodimer interacts with cytoskeleton components, including microtubules, actin and keratin filaments [92]. Phagocytes secrete inflammatory S100A8 and S100A9 proteins, which is induced by their contact with the vascular endothelium, activated under the influence of inflammation [94]. Extracellularly, S100A8 and S100A9 proteins exert many effects through toll-like receptors 4 (TLR4) and RAGE, with which they can bind both as a heterodimer and separately [95,96]. Present in the extracellular space, S100A8 and S100A9 proteins function as molecular structures associated with damage and damage-associated molecular patterns, and affect the activity of vascular endothelial cells as well as epithelial cells, neutrophils, monocytes, macrophages and osteoclasts [97]. 

Many clinical studies suggest an important role of S100B in AD, indicating its increased levels in CSF or the brain of [98,99,100]. In healthy tissues, the S100B protein is present in the largest amounts, for example in astroglial and oligodendroglial cells. In addition, it is expressed in Schwann cells, Müller cells, intestinal glial cells, lining cells of the choroid plexus and some neuronal populations [90]. S100B affects a number of intracellular proteins from the growth factor family, including the growth-associated protein 43 (GAP-43), the regulatory domain of C-kinase, B-cell lymphoma Bcl-2 family proteins and the p53 tumor suppressor protein [88]. The role of S100B is explored as both a potential biomarker and as a pathogenic effector in AD pathology. The effects of the increased concentrations of S100B accompany and contribute to neurodegeneration processes (Figure 2).

In addition to the effect of S100B on RAGE-mediated ROS overproduction, this protein, at concentrations of ≥500 nM, enhances Aβ neurotoxicity. However, at lower concentrations, S100B showed a protective effect against the neurotoxicity of Aβ_25–35_ [101]. Furthermore, recent studies suggested that elevated S100B levels have harmful effects on oligodendrogenesis and myelination through RAGE-dependent processes. On organotypic cerebellar slice cultures, high levels of S100B also compromised neuronal and synaptic integrity, while inducing astrogliosis as well as the activation and inflammation of nuclear factor kappa-light-chain-enhancer of activated B cells. An excessive and prolonged activation of microglia and astrocytes, possibly caused by Aβ, results in the induction of neuronal apoptosis and damage to the blood-brain barrier, which leads to the development of local inflammation and degenerative processes. The hyperactivation of astrocytes may play an important role in brain pathologies, including AD, and S100B may be one of the factors responsible for this process [102]. Triggered microglia produce an increased amount of IL-1, which is also a component of senile plaques. Augmented IL-1 levels, in turn, promote increased secretion of S100B by astrocytes [103]. A S100A9-driven inflammatory cascade may also be involved in the accumulation of Aβ; moreover, it co-aggregates with Aβ_40_ and Aβ_42_ and promotes their amyloid deposition [104]. Isolated astrocyte cultures exposed to micro-molar concentrations of S100B were shown to exhibit the characteristics of a reactive gliosis process by activating substrate pathways 1 of botulinum toxin bound to the Ras protein—the cell division protein 42 C3Rac-1-Cdc42, ERK-Akt and NF-κB. The stimulation of Rac1-Cdc42 proteins led to astroglia growth and increased mitosis. Triggering the ERK-Akt pathway increased survival in oxidative stress conditions. Eminent serum or CSF S100B protein levels are often seen as a marker of glia damage and activation [105]. Hence, measurements of the S100B protein concentration in serum and CSF can potentially be used as a screening parameter and in assessing the progression of damage to the central nervous system. However, the studies on S100B concentrations in CSF and serum in AD cases are partially contradictory [90]. AD victims were found to demonstrate lower serum S100B levels than elderly controls [106]. Another unclear issue is that some studies indicate a correlation between CSF S100B levels and AD brain atrophy or cognitive status [100,107], while others did not find differences between AD cases and controls in regard to their S100B concentrations in CSF [108]. However, the results from a cohort study revealed that a heightened overproduction of S100B is positively correlated with cognitive capacity, especially for perceptual speed and verbal ability in healthy older adults [109]. The secretion of S100B by astrocytes and oligodendrocytes may be augmented in pathological conditions; therefore, its increased concentration, with the concentration of “non-secretory” glial and neuronal proteins, i.e., acid filamentous protein, alkaline myelin protein, and neuro-specific endolase remaining unchanged, may indicate an increased active secretion.

Additional data showing the importance of S100B in neurodegenerative processes are provided by animal studies. Transgenic mice overexpressing S100B and with mutation in APP (Tg2576/S100B) produced worse results in a spatial learning study, which is based on the functioning of the hippocampus. The mice with an inactivated gene encoding S100B showed increased spatial memory and anxiety storage, and increased long-term synaptic enhancement in the hippocampal CA1 sector. This indicates that the S100B extracellular space may play a considerable role in synaptic plasticity [102]. However, research is needed to clarify the relationships and changes in the concentration of this protein so that it can serve as a reliable marker of degenerative changes in AD. Literature data indicate a relationship between glucose concentration and S100B protein secretion. Kheirouri et al. [110] measured levels of S100B protein in the serum of subjects with metabolic syndrome characterized by impaired fasting glucose, central obesity, dyslipidemia and hypertension. The study participants also demonstrated elevated insulin levels and the homeostatic model assessment of Insulin Resistance (HOMA-IR). Serum S100B protein levels were shown to be significantly increased compared to healthy volunteers. In a study by Chuang et al. [111] carried out on mouse mesangial cells 13 (MES-13), it was revealed that under high glucose conditions (27.5 mM), there is an increased expression of six transmembrane epithelial antigens 4 (Steap4) in cell membranes. Under high glucose conditions, a stronger interaction of the S100B protein with Steap4 proteins was also observed. Steap4 proteins reduced the increased expression of inflammatory mediators induced by increased glucose and S100B protein, including transforming growth factor β1 [112]. Astrocyte cultures obtained from rat brains, under metabolic stress conditions, i.e., subjected to glucose, oxygen, and serum deprivation conditions, were followed by an increased secretion of the S100B protein. However, after 12 and 24 h of cell residence under conditions of metabolic stress, expression of S100B mRNA significantly decreased, and after 48 h of incubation, a significant decrease in the S100B protein secretion was observed. Based on these observations, it was hypothesized that the S100B protein could be actively secreted by microglia in the early stages of metabolic stress [113].

### 4.3. Diagnostic and Potential AD Biomarkers from Cerebrospinal Fluid

Three biological markers that can be measured with validated diagnostic tests and can be used in clinical trials as criteria for inclusion and/or determination of the study endpoint have now been approved [114]. These are the concentration of Aβ_42_, total tau (t-tau) and phosphorylated (p-tau) containing phosphorylated Thr181 as they show high accuracy for the differentiation between AD-affected cases and healthy controls. Despite of being pathologically altered years before the onset of the disease, their diagnostic value in detecting MCI subjects due to AD remains controversial [115]. In healthy people, about 50% of total Aβ is Aβ_40_ peptide, about 16%—Aβ_38_ and about 10%—Aβ_42_ peptide. In AD, and especially in the early stages of disease development, Aβ_42_ and Aβ_40_ levels change. In healthy individuals, the concentration of Aβ_40_ in CSF is about 14.6 ng/mL, while in AD sufferers an increase of this value is observed. In healthy individuals, the concentration of Aβ_42_ is in the range of 600–890 pg/mL. In the initial stage of AD, the concentration decreases to 445–614 pg/mL, which can be observed many years before the appearance of clinical symptoms of the disease [116,117]. In the brain of people with AD, Aβ_42_ is the main peptide found in amyloid plaques. In vivo positron emission tomography studies revealed an inverse correlation between Aβ_42_ levels in CSF and accumulation of senile plaques. In people with already observed symptoms of AD, the level of Aβ_42_ in CSF remains relatively stable; therefore, this biomarker is of little importance for monitoring disease progression in these participants [115]. Another analysis of CSF in AD victims shows a significant reduction of Aβ_42_ (<500 pg/mL) in comparison to controls with 794 ± 20 pg/mL [118]. The intraneuronal inclusion of the microtubule-associated protein tau serves as another established biomarker for AD. Tau protein, known to increase gradually with age from <300 pg/mL (21–50 years) to almost <500 (>71 years), demonstrates a significant exponential increase in AD sufferers from >450 to >600 pg/mL (in age rage 51–70), hence proving to be a good prognostic biomarker [119]. AD exhibits the condition of tau protein being phosphorylated in almost 39 possible sites, with position 181 working as a definite biomarker in AD. Other notable phosphorylated tau epitopes include phosphor-tau-199, -231, -235, -396, and -400 [120]. Hyperphosphorylation of the tau protein in the above-mentioned epitopes leads to a decrease in its affinity for microtubules and, consequently, to destabilization of microtubules, disturbance of axonal transport, and neurodegeneration [115]. Phosphorylated epitopes of the tau protein include Thr181, Thr231, Thr235 and Ser199, Ser396, Ser404. 

There are also studies aimed to classify other indicators under the group of CSF biomarkers. The ratio of total tau/Aβ_42_ and the level and ratio of phosphorylated tau/Aβ_42_ found application in predicting the progression of mild cognitive impairment to AD and in people with normal cognitive functions. In addition, the European Medicines Agency approved the total tau/Aβ_42_ ratio as an additional biomarker in research on γ-secretase inhibitors that could be used in AD treatment [121]. The Aβ_38_/Aβ_42_ ratio, β-secretase 1 levels, monocyte chemoattractant protein-1 (MCP1), cardiac fatty acid binding protein (hFABP) concentration are also being tested. BACE1 digests APP protein at the first asparagine, maintaining the structure of Aβ, and its elevated levels in CSF were detected in participants with AD [122]. The hFABP protein is primarily found in cardiomyocytes and skeletal muscle cells, and in smaller amounts in a healthy brain and it participates in the metabolism and intracellular transport of fatty acids. An increased concentration of hFABP was found in subjects with AD, both in the initial stages of the disease and in its further stages [123]. Also, plasma levels of inflammatory mediators such as monocyte chemoattractant protein-1 (MCP1)—combined with the analysis of cognitive decline with Mini-Mental state examination (MMSE)—have prognostic potential [124]. It was discovered that plasma MCP-1 concentrations tend to increase along with the deteriorating AD state, which was reflected in worsening results of MMSE. 

### 4.4. Diagnostic and Potential Biomarkers from Peripheral Blood

The filtering mechanism of the BBB prevents the diffusion of substances into blood; thus, the detection of blood biomarkers exhibits lower sensitivity and specificity than that of biomarkers detectable in participants’ CSF. The most studied AD biomarker measured in peripheral blood is Aβ. The concentration of Aβ in peripheral blood is much lower than in CSF, which can cause analytical and methodological difficulties. Historically, the ratio of plasma Aβ_42_ to Aβ_40_ was used to optimize the concordance between CSF levels and positron-emission tomography. There was a positive correlation between Aβ_40_ and Aβ_42_ in peripheral blood and in CSF. Concentrations of Aβ_40_ and Aβ_42_ in peripheral blood in victims of AD were reduced compared to other study groups. Aβ_40_ and Aβ_42_ levels are not validated peripheral blood AD biomarkers, but may potentially be valuable screening biomarkers [125,126]. Aβ_40_ is the dominant form of Aβ peptide in the brain, CSF and plasma. Meta-analysis of cohort studies revealed only a weak relationship between its concentration and AD. In addition, a decreased blood Aβ_40_/Aβ_42_ ratio was associated with the risk of developing mild cognitive impairment (MCI) or dementia in people with normal cognitive function [115,126]. Other studies point to varied forms of amyloid beta in blood plasma as crucial potential research into AD biomarkers. For example, it was shown that the levels of Aβ_17_ in plasma samples of MCI and age-matched control groups vary significantly. The concentrations of this Aβ form might be a highly sensitive and specific biomarker for AD as it is the second most abundant CSF Aβ form [127]. The tau protein and neurofilament light chains (NFL) are also candidates for blood screening biomarkers. However, these are non-specific indicators. Their increased concentration in blood indicates neurodegeneration processes not only associated with AD [128,129]. Ray et al. [130] examined a variety of plasma-based proteins and developed an algorithm that accurately classified AD based on the measurement of 18 plasma signaling protein profiles including alpha-1-antitrypsin, complement factor H, alpha-2-macroglobulin, apolipoprotein J, apolipoprotein A-I. That study was followed by another analysis of serum-protein multiplex biomarkers with a smaller number of serum proteins and including clinical labs [131]. Studies on the potential use of Aβ_40_ and Aβ_42_ concentrations in plasma as neuropathological markers are ambiguous. Some studies found reduced concentrations of these two forms of amyloid beta, while others demonstrated an increase of peptide levels in participants with AD compared to non-demented cases. Recently, studies analyzing cerebral Aβ by detecting plasma Aβ_42_, Aβ_40,_ tau levels together with NFL and neurofilament heavy chain (NFH) and the *APOE* genotype have been performed. As the reference standard for brain Aβ levels, they used a CSF Aβ_42_/Aβ_40_ ratio validated against the amyloid PET status. Those authors revealed that the plasma amyloid ratio predicts the Aβ status in all stages of AD with similar accuracy in a validation cohort [132]. Lately, there is a growing body of evidence that the measurement of NFL concentrations in plasma might be considered a potential marker for neurodegenerative changes characteristic of AD. Lewczuk et al. identified elevated NFL levels in cases with AD compared to healthy controls, which were inversely correlated with MMSE results [133]. 

## 5. Treating AD with Diabetic Medications 

The very interesting and quite new approach to the comorbidity of different diseases is based on the observation that each subject and disease entity is heterogeneous in the profile and need a multidirectional approach. It seems that there are disease-associated genes, protein–protein interactions, metabolic relationships or signaling pathways which are shared between distinct illnesses. Even though broadly understood glucose homeostasis disturbances are currently intensively examined, the manifestation of AD across different individual might be also different and does not have to relate only to T2DM. Among subjects with AD different rates of progression have been observed. Risk factors that accelerate deterioration have been identified and some of them, such as genetics, comorbidity, and the early appearance of Alzheimer disease motor signs are indicated [134]. It was shown that analyzing the overlap in the direct adjacent of the disease genes as well as overlay of the biological pathways helps in better understanding of the interplay between diseases and their relationships [135]. This type of analyses and approach could be useful in designing new multi-directional drugs or in re-designing old drugs for new purposes. The new polypharmacology trend is to create drugs that will target common biological pathways involved in more than one disease. In this light a new view at the old use of some drugs, especially this applied in the T2DM treatment seems interesting. The drug repurposing candidates that can target common mechanisms involved in T2DM and AD have recently gained attention due to the increased comorbidity among such cases [136]. In the paper of Aguirre-Plans et al. [137] authors proposed a novel drug repurposing approach, Proximal pathway Enrichment Analysis”—a powerful computational strategy for targeting multiple pathologies involving common biological pathways in T2DM and AD. 35 of evaluated pathways are shared by both disease entities. Using the new method authors have proposed Orlistat, a well-known anti-obesity and T2DM drug for potential anti-AD medication. Several proteins connected to T2DM-AD interrelation such as apolipoprotein A1, presenilin 2, pancreatic lipase, lipoprotein lipase, and Ig heavy constant γ1, are Orlistat’s targets or in the proximity of the targets. 

Due to specially intimate relationship between brain insulin resistance and AD, and the relative “paralysis” of brain insulin signaling through the IRS-1 → AKT pathway in mild MCI and AD, there are voices suggesting that restoring signaling through this pathway with therapeutic agents originally developed for the treatment of T2DM, may be of particular benefit. As especially important, exogenous insulin. Other therapies, including GLP-1 agonists (e.g., exenatide, liraglutide), metformin, leptin analogs (metreleptin), amylin analogs (pramlintide), and protein–tyrosine phosphatase 1B (PTP1B) inhibitors may circumvent insulin-signaling impairment and reestablish signaling through this pathway (IRS-1 → AKT). The smaller role is attributed to peroxisome proliferator–activated receptor-γ agonists (e.g., rosiglitazone and pioglitazone) but further study are needed [138]. Repurposing anti-diabetic agents to prevent insulin resistance in AD has gained substantial attention due to the therapeutic potential it offers. The 6-h insulin infusion in non-impaired adults improved cognitive abilities as well as in AD victims where the acute administration improved story recall and attention [139,140] The intranasal insulin delivery, not causing systemic hypoglycemia, improved memory functions in in mild cognitive impairment and AD subjects [141]. In preclinical studies, thiazolidinediones a class of oral T2DM drugs, improved memory, and modulation of Aβ levels in CSF [142]

McIntosh et al. [143] in large group of AD subjects with different stages of glucose disturbances examined relationships between treatment status and CSF biomarkers and risk for dementia. Subjects were grouped by fasting blood glucose and medication history: Euglycemia (EU), pre-T2DM (PD), untreated T2DM (UD), and treated T2DM (TD). It was one or more antidiabetic medication (biguanides, sulfonylureas, thiazolidinedione, dipeptidyl peptidase inhibitor, glucagon like peptide-1 agonist, insulin). They observed/revealed that the UD group displayed significantly greater p-tau, t-tau, and p-tau/Aβ_42_ levels than the EU, PD, and TD groups and higher t-tau/Aβ_42_ than the EU and PD groups). The UD group progressed to dementia at higher rates than the EU group (hazard ratio 1.602 (95% CI 1.057–2.429); *P* = 0.026). Authors concluded that treatment status may alter the relationship between T2DM and both AD biomarker profile and risk of dementia, and UD is associated with elevated tau pathology and risk of dementia, whereas TD is not. This support the potential importance of treatment status in AD risk associated with T2DM and create this as an important aspect in management of AD sufferers. 

Some controversy concerns insulin therapy. Plastino et al. [144] evaluated cognitive impairment in patients with AD and T2DM treated with either oral antidiabetic drugs or combination of insulin with other T2DM medications during 1 year observation study. Subjects with mild-to-moderate AD and T2DM were divided into two groups, according to antidiabetic pharmacotherapy: Patients treated with oral antidiabetic drugs (group A), and patients treated with insulin combined with other oral antidiabetic medications (group B). At the end of the study the significant worsening of cognitive functions in more than half of the patients from group A and above more than a quarter of patients of group B, compared to baseline MMSE scores. However ischemic heart disease and hypertension were significantly higher in group B. But finally authors suggests that insulinic therapy could be effective in slowing cognitive decline in cases with AD. Similarly, Alagiakrishnan et al. [145] indicates that T2DM therapies may improve cognitive function in AD patients, especially intranasal insulin has been shown to improve memory and cognitive abilities in mild cognitive impairment and AD patients.

Lately it was indicated that poor lifestyle, age, hyperglycemia, hypercholesterolemia, and inflammation are some of the major risk factors that contribute to cognitive and memory impairments in T2DM sufferers. Sharma et al. [146] studies showed that physical inactivity, frequent alcohol consumptions, and use of packed food products that provides an excess of cheap calories are found associated with cognitive impairment and depression in T2DM cases, so its modification can be interpreted as an important gripping point in non-conventional therapeutic action. Authors suggest that omega fatty acids (FAs) and polyphenol-rich foods, especially flavonoids, can reduce the bad effects of an unhealthy lifestyle and can modify such agents, therefore, the consumption of omega FAs and flavonoids may be beneficial in maintaining normal cognitive function especially in T2DM cases. The application of healthy nutrients and foods such as functional foods may have potential health benefits and may act as promising therapeutic agents both in T2DM and subjects with cognitive decline and AD. The functional foods act by various modes in the cells chiefly by enhancing antioxidant activity, anti-inflammatory, cellular signaling, improved insulin sensitivity, and reduced insulin resistance.

## 6. Conclusions

The underlying biological mechanisms that link the development of T2DM with AD are not fully understood. Abnormal protein processing, accumulation of amyloid plaques and tau protein abnormalities, the reduction in acetylcholine, dyslipidemia and hypercholesterolemia, oxidative stress and mitochondrial dysfunction, activation of inflammatory pathways are connected with AD pathogenesis but these futures are also indicated as common to both diseases—AD and T2DM. The comorbidity of Alzheimer’s disease and T2DM is an ongoing topic of interest in various studies, especially in the context of potential AD biomarkers given that these diseases have common pathophysiology. Changes in glucose metabolism observed in the early- and late-onset form of AD suggest a promising field for experiments in this issue, as well as a further examination of this link.

## Figures and Tables

**Figure 1 ijms-21-02744-f001:**
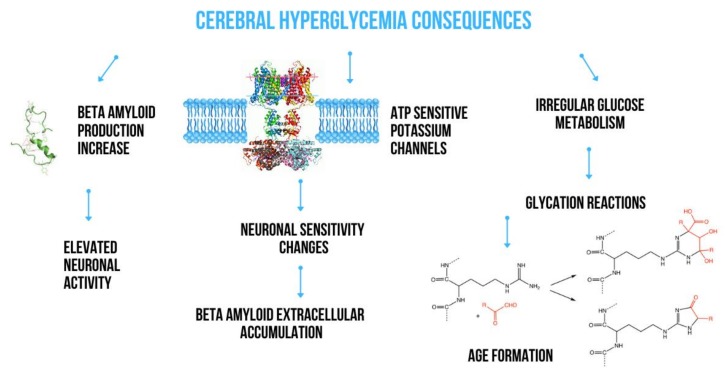
Consequences of acute hyperglycemia action in the brain.

**Figure 2 ijms-21-02744-f002:**
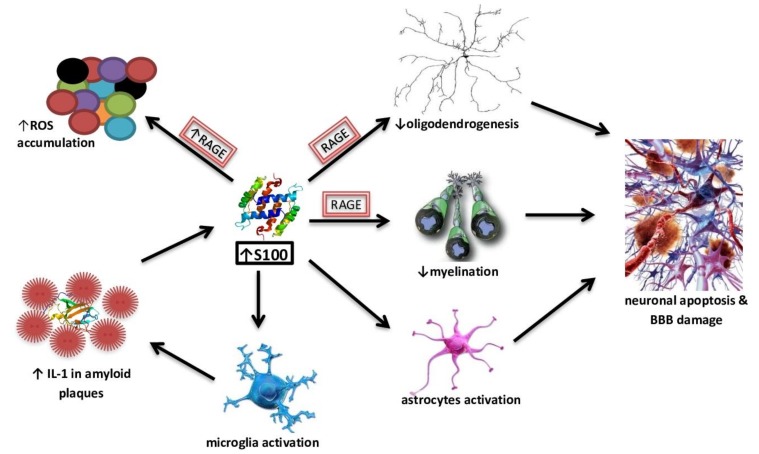
The effect of S100B high concentrations on neurodegeneration processes.

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
