# Peer review of "The Interplay between Diabetes and Alzheimer’s Disease—In the Hunt for Biomarkers"

_ijms, 2020, doi:10.3390/ijms21082744_

Round 1
Reviewer 1 Report
The review by Kubis-Kubiak et al on the Interplay between type 2 diabetes (T2D) and Alzheimer’s
Disease (AD) is a timely and considerably comprehensive. I have several suggestions towards the improvement of the text:
1. In my opinion it is important to acknowledge various hypotheses in regards to the pathology giving rise to AD such as the reduction in acetylcholine, accumulation of amyloid plaques and tau protein abnormalities. A brief text mentioning these hypotheses in the introduction would help to provide context to the reader.
2. In parallel with my earlier comment and under the light of advancements in the field of precision medicine, it would be beneficial to reader to discuss that the manifestation of AD across different individual might be different and does not have to relate to T2D.
3. The text lacks few references relevant for the comorbidity and common disease mechanisms underlying AD and T2D.
Sims-Robinson, C.; Kim, B.; Rosko, A.; Feldman, E.L. How does diabetes accelerate Alzheimer disease pathology? Nat. Rev. Neurol. 2010, 6, 551–559.
Hiltunen, M.; Khandelwal, V.K.M.; Yaluri, N.; Tiilikainen, T.; Tusa, M.; Koivisto, H.; Krzisch, M.; Vepsäläinen, S.; Mäkinen, P.; Kemppainen, S.; et al. Increased risk of type 2 diabetes in Alzheimer disease. J. Cell. Mol. Med. 2012, 16, 1206–1222.
4. Related to my previous comment, several approaches have been developed in the recent years towards characterizing endophenotypes and comorbidities between AD and T2D. In fact, some of these studies suggest the use of T2D drugs in AD. I feel the discussion of such computational methods and drug repurposing would be a valuable addition to the review.
Duthie A.; Chew D.; Soiza R.L. Non-psychiatric comorbidity associated with Alzheimer’s disease. QJM: An International Journal of Medicine, Volume 104, Issue 11, November 2011, 913–920.
Yarchoan, M.; Arnold, S.E. Repurposing Diabetes Drugs for Brain Insulin Resistance in Alzheimer Disease. Diabetes 2014, 63, 2253–2261.
Rubio-Perez, C.; Guney, E.; Aguilar, D.; Piñero, J.; Garcia-Garcia, J.; Iadarola, B.; Sanz, F.; Fernandez-Fuentes, N.; Furlong, L.I.; Oliva, B. Genetic and functional characterization of disease associations explains comorbidity. Sci. Rep. 2017, 7, 6207.
Aguirre-Plans, J.; Pinero, J.; Menche, J.; Sanz, F.; Furlong, L.I.; Schmidt, H.; Oliva, B.; Guney, E. Proximal Pathway Enrichment Analysis for Targeting Comorbid Diseases via Network Endopharmacology. Pharmaceuticals 2018, 11, 61.
Author Response
Dear Reviewer I
Below You can find my responses to your annotations. I hope that you will be pleased with changes that I have made.
best reagards,
REVIEWER I:
The review by Kubis-Kubiak et al on the Interplay between type 2 diabetes (T2D) and Alzheimer’s
Disease (AD) is a timely and considerably comprehensive. I have several suggestions towards the improvement of the text:
- In my opinion it is important to acknowledge various hypotheses in regards to the pathology giving rise to AD such as the reduction in acetylcholine, accumulation of amyloid plaques and tau protein abnormalities. A brief text mentioning these hypotheses in the introduction would help to provide context to the reader.
Ad. The paragraph 1.1 Alzheimer Disease characteristic was added from line 32
- In parallel with my earlier comment and under the light of advancements in the field of precision medicine, it would be beneficial to reader to discuss that the manifestation of AD across different individual might be different and does not have to relate to T2D.
Ad. The paragraph was added from line 180
- The text lacks few references relevant for the comorbidity and common disease mechanisms underlying AD and T2D.
Sims-Robinson, C.; Kim, B.; Rosko, A.; Feldman, E.L. How does diabetes accelerate Alzheimer disease pathology? Nat. Rev. Neurol. 2010, 6, 551–559.
Hiltunen, M.; Khandelwal, V.K.M.; Yaluri, N.; Tiilikainen, T.; Tusa, M.; Koivisto, H.; Krzisch, M.; Vepsäläinen, S.; Mäkinen, P.; Kemppainen, S.; et al. Increased risk of type 2 diabetes in Alzheimer disease. J. Cell. Mol. Med. 2012, 16, 1206–1222.
Ad. The paragraph was added in line 94 and 193.
- Related to my previous comment, several approaches have been developed in the recent years towards characterizing endophenotypes and comorbidities between AD and T2D. In fact, some of these studies suggest the use of T2D drugs in AD. I feel the discussion of such computational methods and drug repurposing would be a valuable addition to the review.
Duthie A.; Chew D.; Soiza R.L. Non-psychiatric comorbidity associated with Alzheimer’s disease. QJM: An International Journal of Medicine, Volume 104, Issue 11, November 2011, 913–920.
Yarchoan, M.; Arnold, S.E. Repurposing Diabetes Drugs for Brain Insulin Resistance in Alzheimer Disease. Diabetes 2014, 63, 2253–2261.
Rubio-Perez, C.; Guney, E.; Aguilar, D.; Piñero, J.; Garcia-Garcia, J.; Iadarola, B.; Sanz, F.; Fernandez-Fuentes, N.; Furlong, L.I.; Oliva, B. Genetic and functional characterization of disease associations explains comorbidity. Sci. Rep. 2017, 7, 6207.
Aguirre-Plans, J.; Pinero, J.; Menche, J.; Sanz, F.; Furlong, L.I.; Schmidt, H.; Oliva, B.; Guney, E. Proximal Pathway Enrichment Analysis for Targeting Comorbid Diseases via Network Endopharmacology. Pharmaceuticals 2018, 11, 61.
Ad. The above mentioned references were incorporated in the paper mostly in the newly created paragraph 5. Treating AD with diabetic medications.
The whole paper has been rebuild, shortened and more clearly divided into new subsections. The paragraph describing S100 protein family has been incorporated in the paragraph with biomarkers.
Reviewer 2 Report
This review article addresses an interesting and important topic. However, there are several edits that would improve the manuscript.
Line 13: refers to toxic effects of glucose - consider replacing with detrimental.
Line 15: blood glucose “near limits” – this is a vague phrase. Specify - what limits?
Line 39: consider replacing “absorbtion” with “transport”
Line 59: do the authors mean “granules” not grains?
Line 74: proven is too strong of a word – consider using shown
General: throughout the manuscript, authors refer to participants in the studies as “patients.” They should consider changing this to “participants” or “subjects” as patient suggests that these individuals are being clinically treated.
Line 93: it is unclear what the sentence “Moreover, elevated glucose levels are associated with regions vulnerable to the accumulation of […]” Associated with what about these regions?
Line 95: again, proven is too strong of a word. Evidence has accumulated to suggest…
Line 98: A-beta accumulation in what tissue? (in T2D – also, need to add a citation). While there is certainly evidence for a relationship between amyloid beta and insulin (for instance, through IDE) there is some clinical evidence that cerebral neuropathology, at least, may be lower in individuals with T2D (see Thambisetty et al 2013)
Line 332: < and > seem to be switched for the number 500 – the sentence on this line and the following (line 333) is confusing.
Neurofilament light chain is mentioned on line 379 and not defined until 394
MMSE not defined. A number of acronyms, such as type 2 diabetes, Alzheimer’s Disease, etc vary between being spelled out and abbreviated throughout the manuscript.
S100 protein family paragraph is not introduced well - need to seed/introduce this concept.= including roles/relevance to AD and brain health earlier.
Author Response
Dear Reviewer II
Below You can find my responses to your annotations. I hope that you will be pleased with changes that I have made.
best reagards,
This review article addresses an interesting and important topic. However, there are several edits that would improve the manuscript.
Line 13: refers to toxic effects of glucose - consider replacing with detrimental.
Ad. The word “toxic” was replaced with “detrimental”
Line 15: blood glucose “near limits” – this is a vague phrase. Specify - what limits?
Ad. The phrase “blood glucose near limits…” was replaced with “blood glucose levels near lower fasting limits (72 to 99 mg/dL)…”
Line 39: consider replacing “absorbtion” with “transport”
Ad. The word “absorbtion” was replaced with “transport”
Line 59: do the authors mean “granules” not grains?
Ad. The word “grain” was replaced with “granules”
Line 74: proven is too strong of a word – consider using shown
Ad. The word “proven” was replaced with “shown”
General: throughout the manuscript, authors refer to participants in the studies as “patients.” They should consider changing this to “participants” or “subjects” as patient suggests that these individuals are being clinically treated.
Ad. The words “patients” were replaced with: “subjects”, “participants”, “sufferers “, “victims” and “cases”.
Line 93: it is unclear what the sentence “Moreover, elevated glucose levels are associated with regions vulnerable to the accumulation of […]” Associated with what about these regions?
Ad. The phrase “Moreover, elevated glucose levels are associated with regions vulnerable to the accumulation of Aβ and NFTs” was replaced with “Moreover, elevated levels of brain tissue glucose are associated with greater severity of both Aβ deposition and neurofibrillary pathology.”
Line 95: again, proven is too strong of a word. Evidence has accumulated to suggest…
Ad. The phrase “…the brain has been proven as an insulin target region…” has been replaced with “…evidence has accumulated to suggest that the brain could be an insulin target region…”
Line 98: A-beta accumulation in what tissue? (in T2D – also, need to add a citation). While there is certainly evidence for a relationship between amyloid beta and insulin (for instance, through IDE) there is some clinical evidence that cerebral neuropathology, at least, may be lower in individuals with T2D (see Thambisetty et al 2013)
Ad. The adequate paragraph was added from line 180.
Line 332: < and > seem to be switched for the number 500 – the sentence on this line and the following (line 333) is confusing.
Ad. I don’t understand this annotation. There is no [and] word in line 332 and it’s hard to find to which [and] do You refer in the proximity of line 332. Please, if you could describe more clearly what you meant.
Neurofilament light chain is mentioned on line 379 and not defined until 394
Ad. The words “neurofilament light chain” was added in line 589 and cancelled from line 601
MMSE not defined. A number of acronyms, such as type 2 diabetes, Alzheimer’s Disease, etc vary between being spelled out and abbreviated throughout the manuscript.
Ad. The MMSE abbreviation has been defined in line 569. The words ”type 2 diabetes” or “diabetes“ was replaced with abbreviation “T2DM”. The words ”Alzheimer’s disease” was replaced with abbreviation “AD”.
S100 protein family paragraph is not introduced well - need to seed/introduce this concept.= including roles/relevance to AD and brain health earlier.
Ad. The whole paper has been rebuild, shortened and more clearly divided into subsections according to the list below:
- Alzheimer’s Disease vs Type 2 Diabetes Mellitus
1.1 Alzheimer Disease characteristic
1.2 Carbohydrate Metabolism
1.2.1 Glucose Metabolism in the Brain
1.2.2 Insulin Role in the Brain
1.3 Brain Energy Metabolism and AD onset
2.The Effect of Metabolic Disturbances on Amyloid Production
2.1 Impact of Glucose Metabolism on Amyloidogenesis
2.2. The Role of Insulin Signaling in the Amyloid β Cascade
3.The Effect of Metabolic Disturbances on tau protein metabolism
3.1 Impact of Glucose Metabolism on Tau Protein Aggregation
3.2 Improper Insulin Metabolism and Tau Protein Synthesis
- Biomarkers for Alzheimer’s Disease Diagnosis
4.1. T2DM-Related Biomarkers in AD
4.2 S100 Protein Family as a potential biomarkers of AD
4.3. Diagnostic and Potential AD Biomarkers from Cerebrospinal Fluid
4.4. Diagnostic and Potential Biomarkers from Peripheral Blood
- Treating AD with T2DM medications
- Conclusions
The paragraph describing S100 protein family has been incorporated in the paragraph with biomarkers. The whole new paragraph 5. Treating AD with diabetic medications was created.
Reviewer 3 Report
This is a review article regarding the interplay between diabetes and Alzheimer’s disease (AD). The authors firstly describe overview of the interaction between diabetes and AD, i.e., hyperglycemia and insufficient insulin action in the brain, and then describe the biomarker of AD, paying attention to S100B protein.
- Overall, the review was superficial and catalog-like and it is difficult to follow. There should be more connection between sections 1 and 2.
- Each paragraph is too long and difficult to read. It should be shortened.
- Figures and tables are too simple and they do not help to understand the contents of the review.
- If the authors want to focus on S100B protein, the description of S100B should be positioned as the center of the review.
Author Response
Dear Reviewer III
Below You can find my responses to your annotations. I hope that you will be pleased with changes that I have made.
best reagards,
This is a review article regarding the interplay between diabetes and Alzheimer’s disease (AD). The authors firstly describe overview of the interaction between diabetes and AD, i.e., hyperglycemia and insufficient insulin action in the brain, and then describe the biomarker of AD, paying attention to S100B protein.
Overall, the review was superficial and catalog-like and it is difficult to follow. There should be more connection between sections 1 and 2. Each paragraph is too long and difficult to read. It should be shortened.
Ad. The whole new paragraph 5. Treating AD with diabetic medications was created. The whole paper has been rebuild, shortened and more clearly divided into new subsections according to the list below:
- Alzheimer’s Disease vs Type 2 Diabetes Mellitus
1.1 Alzheimer Disease characteristic
1.2 Carbohydrate Metabolism
1.2.1 Glucose Metabolism in the Brain
1.2.2 Insulin Role in the Brain
1.3 Brain Energy Metabolism and AD onset
2.The Effect of Metabolic Disturbances on Amyloid Production
2.1 Impact of Glucose Metabolism on Amyloidogenesis
2.2. The Role of Insulin Signaling in the Amyloid β Cascade
3.The Effect of Metabolic Disturbances on tau protein metabolism
3.1 Impact of Glucose Metabolism on Tau Protein Aggregation
3.2 Improper Insulin Metabolism and Tau Protein Synthesis
- Biomarkers for Alzheimer’s Disease Diagnosis
4.1. T2DM-Related Biomarkers in AD
4.2 S100 Protein Family as a potential biomarker of AD
4.3. Diagnostic and Potential AD Biomarkers from Cerebrospinal Fluid
4.4. Diagnostic and Potential Biomarkers from Peripheral Blood
- Treating AD with T2DM medications
- Conclusions
Figures and tables are too simple and they do not help to understand the contents of the review.
Ad. The new figure 2 was created. The old figures 2, 3 and 4 with tables has been cancelled.
If the authors want to focus on S100B protein, the description of S100B should be positioned as the center of the review.
Ad. The paragraphs with S100 protein family and potential role of S100B protein in AD diagnosis were positioned before the description of other biomarkers in CSF and blood.
Round 2
Reviewer 3 Report
The auhtors have responded to the comments appropriately.